# Humans as geomorphic agents: Lidar detection of the past, present and future of the Teotihuacan Valley, Mexico

Nawa Sugiyama[1]*, Saburo Sugiyama[2,3], Tanya Catignani[4], Adrian S. Z. Chase[5], Juan C. Fernandez-Diaz[6]

1 Department of Anthropology, University of California-Riverside, Riverside, CA, United States of America, 2 School of Human Evolution and Social Change, Arizona State University, Tempe, AZ, United States of America, 3 Research Institute for the Dynamics of Civilizations, Okayama University, Kita-ku, Okayama, Okayama, Japan, 4 Department of Sociology & Anthropology, George Mason University, Fairfax, VA, United States of America, 5 Advanced GIS Lab, Center for Information Systems & Technology, Claremont Graduate University, Claremont, CA, United States of America, 6 National Center for Airborne Laser Mapping (NCALM) and Department of Civil and Environmental Engineering, University of Houston, Houston, TX, United States of America

* nawa.sugiyama@ucr.edu

**Data Availability Statement:** Quantitative data is reported in the supplementary information. The lidar data is part of Mexico's cultural patrimony and contains highly sensitive information including the

## Abstract

As humans are the primary geomorphic agents on the landscape, it is essential to assess the magnitude, chronological span, and future effects of artificial ground that is expanding under modern urbanization at an alarming rate. We argue humans have been primary geomorphic agents of landscapes since the rise of early urbanism that continue to structure our everyday lives. Past and present anthropogenic actions mold a dynamic "taskscape" (not just a landscape) onto the physical environment. For example, one of the largest Pre-Columbian metropolitan centers of the New World, the UNESCO world heritage site of Teotihuacan, demonstrates how past anthropogenic actions continue to inform the modern taskscape, including modern street and land alignments. This paper applies a multi-scalar, *long durée* approach to urban landscapes utilizing the first lidar map of the Teotihuacan Valley to create a geospatial database that links modern and topographic features visible on the lidar map with ground survey, historic survey, and excavation data. Already, we have recorded not only new features previously unrecognized by historic surveys, but also the complete erasure of archaeological features due to modern (post-2015) mining operations. The lidar map database will continue to evolve with the dynamic landscape, able to assess continuity and changes on the Teotihuacan Valley, which can benefit decision makers contemplating the stewardship, transformation, or destruction of this heritage landscape.

## Introduction

We live in the Anthropocene, where humans are principal agents conditioning environmental and climatic processes on earth. Geologists and archaeologists recognize anthropogenic alterations cause global changes and that deep-time human environmental impacts prohibit

exact location of archaeological features that may attract looters. Researchers seeking access to this resource must contact the Instituto Nacional de Antropología e Historia (INAH consejo. arqueologia@gmail.com) which is currently establishing guidelines for the dissemination of lidar data in Mexico consistent with the protection archaeological sites.

**Funding:** Elaboration of the lidar map and field survey was funded by the Japan Society for the Promotion of Science (JSPS, 25257016, 17H01650, 19H05732) through Aichi Prefectural University and the National Science Foundation (Archaeology BCS 1638525).

**Competing interests:** The authors have declared that no competing interests exist.

evaluation of "pristine" landscapes [1–3]. Wilkinson [4], for example, estimated that human geologic agents denude the earth's surface at roughly ten times the combined rate of glaciers, rivers, and other natural processes. In this paper we present a multi-scalar, *long-dureé* approach to recording the Teotihuacan Valley, the site of a flourishing metropolis from CE 1–550 and one of the most anthropogenically altered landscapes in the ancient New World. We apply lidar technology to detect, protect, and archive one of humanity's most numinous feats of environmental alteration: the ceremonial production of an enduring urban landscape. We argue that archaeologists are uniquely situated to contribute valuable insight into the continuity between past and present landscapes [5–7].

Every year, millions of visitors are drawn to Teotihuacan's breathtaking ritual precinct. The grand axial spine of this UNESCO world heritage site, known as the Avenue of the Dead, is flanked by the Moon Pyramid, the Sun Pyramid, and the Ciudadela (Citadel) that encloses the Feathered Serpent Pyramid; staggered silhouettes with a command of the eye despite depredations of nearly two millennia which have passed since their foundations were laid. Handsome and severe, their perpetual testimony to the power of state monumentalism resonates with researchers, tourists, and locals alike. The lidar map of the Teotihuacan Valley attests to the scale and perseverance of the ancient cityscape onto the modern terrain (Fig 1). As a digital archive of the Teotihuacan landscape, its broader impact has already become clear, for at just 45 km NE of Mexico City this ancient urban center is acutely vulnerable to the encroachment and resource demands of its sprawling modern neighbor. When large swathes of the valley were subject to industrial bedrock mining for the aborted construction of an international airport, some 205 archaeological features disappeared by these operations were preserved only in the original 3D snapshot of lidar points taken in 2015 (Fig 2, S1A Table in S1 File).

The Teotihuacan Valley's unique environmental, cultural, and academic trajectories support a multi-scalar definition of humans as geomorphic agents. Lidar sampling resolution of this semi-arid volcanic cordillera is exceptional across the entire region. As a cultural touchstone, this area has been subject to intensive and extensive excavation and survey for over a century, beginning with the exploration and inauguration of the archaeological park for the centennial celebration of the Mexican Independence by Porfirio Diaz in 1910 [8, 9]. However, the exponential growth of adjacent settlements along with the introduction of mechanized agriculture and extensive surface mining complicates the task of distinguishing ancient features obfuscated by this second modern wave of anthropogenic alterations. Project Plaza of the Columns Complex draws on the deep history of archaeological research combined with ground reconnaissance and excavations to interpret the lidar data.

## Teotihuacan

Three principal traits qualify Teotihuacan as a quintessential comparative subject for developing models of urbanism and its long-term legacies on the terrain. 1) The site's total archaeological dossier, representing over a century of exploration including landmark ground surveys, has generated richly layered synchronic and diachronic datasets to map settlement patterns with good spatiotemporal control [10–14]. 2) The size, planning, and cosmopolitan nature of the city all have close analogs in the modern world [15–18]. 3) Mexico City, a mere 45 km away from the site, poses an existential threat to its own archaeological heritage.

The configuration of scale, density, and orthogonal cityscape at Teotihuacan—unparalleled in the New World—supported an estimated 100,000 inhabitants structured into well-defined multi-ethnic and specialized neighborhoods across an area of roughly 20 km$^2$ [19–22]. Every aspect of urban design—from structure orientation (15.5˚ east of astronomical north), raw material sourcing (andesite and plaster), architectural form, and even artistic expression in

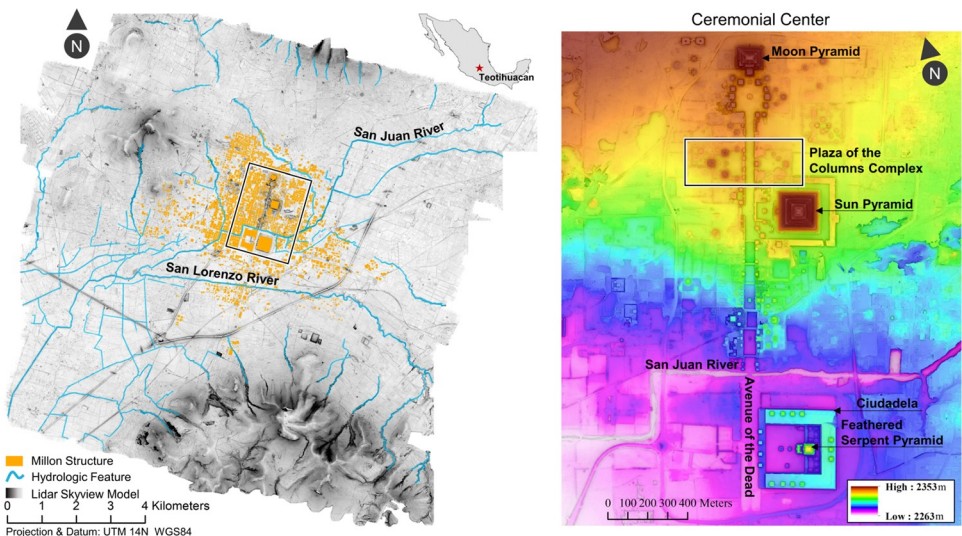

**Fig 1. Lidar maps of Teotihuacan Valley.** A) skyview factor visualization indicating hypothetical city reconstruction by Millon (29) (yellow) and hydrological systems (blue), and B) color-ramped DEM view of the ceremonial center (rectangle box on left map). Published under a CC BY license, with permission from N. Sugiyama, original copyright Project Plaza of the Columns Complex 2021.

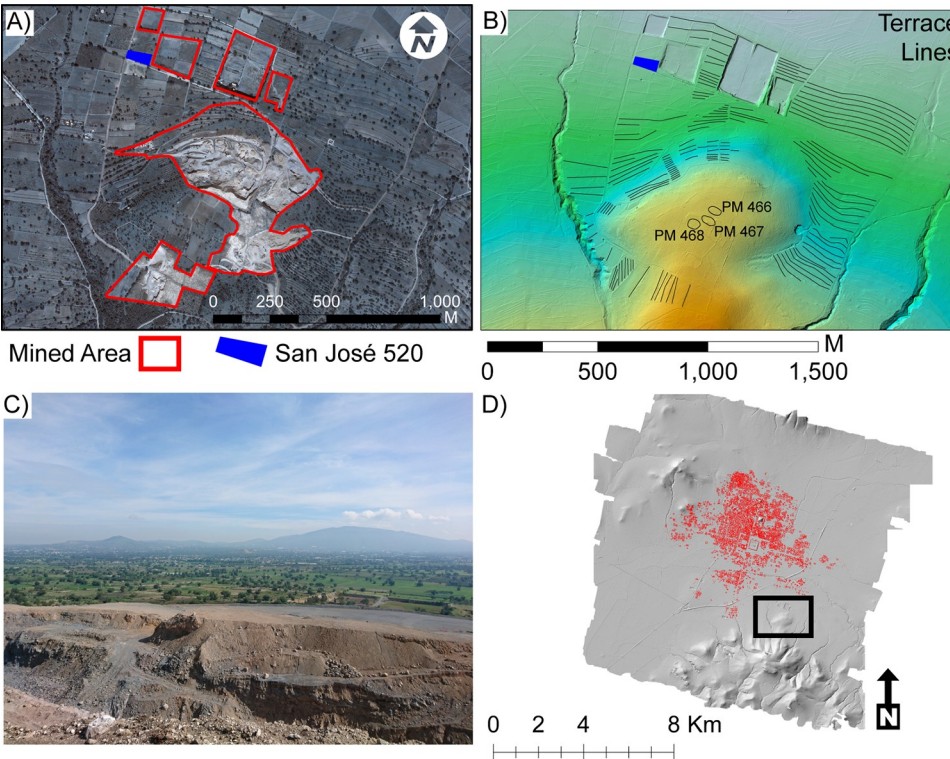

**Fig 2. Mining example from Coatlán, Sierra Patlachique's northern slope.** A) satellite image of mine extent (Worldview-3, December 12, 2017 courtesy of Maxar Technologies), B) lidar color-ramped DEM draped over sky-view factor of the same area from 2015 indicating terrace lines and three potential mounds, C) photograph of the mine taken from ground verifiction (looking southward) (Photograph: A. Texis), D) location of the mine within the Teotihuacan Valley indicated by the rectangle. PM = potential mound.

their mural art—was standardized and thus likely regulated by the state [23–25]. Eighty percent of the population in the Basin of Mexico was concentrated within the urban core of Teotihuacan, making the city an outstanding urban model for both its time and ours [26–28]. However, we have yet to fully comprehend the total ecological footprint of Teotihuacan. Two seminal foot surveys characterized the city; an intensive survey of the city's core that produced a two-dimensional topographic map at 1:2000 m scale overlaid with hypothetical reconstructions of edifices (Teotihuacan Mapping Project, TMP) [29, 30], and an extensive survey of the periphery that contextualized the Teotihuacan Valley within the greater Basin of Mexico (Teotihuacan Valley Project, TVP) [12]. A three-hundred-meter zone absent of structures, ceramic debris, or other evidence of Teotihuacan occupation defined our current understanding of the city's boundary [29]. The researchers mention that insufficient resources precluded the survey of a 4 km$^2$ area to the northwestern extension of Teotihuacan despite the presence of early (pre-Tzacualli) structures therein. Additional Teotihuacan sites reported by the Teotihuacan Valley Project [12] and other excavations in the periphery [31–33] confirm the need to reassess the city's full extent.

Rural-urban transactions greatly inform our understanding of how urban landscapes are socially and economically sustained [34, 35]. Teotihuacan's ecological footprint needs to be evaluated by defining the complex relationship between a city and the hinterland regions which supplied it with food, fuel, building material, and constant migrant labor inflow [12, 16, 18, 21, 36–38]. Many geomorphological and paleobotanical studies in the region confirm extensive anthropogenic erosion and deforestation since the Teotihuacan period [39–42]. Intensive agriculture, mining, and unmanaged exploitation of forest resources (for fuel) induced erosion and significantly changed local hydrologic conditions.

Teotihuacan lies just within the limit of rainfall agricultural capability [43–46] and is exposed to a high risk of seasonal flooding. Hydrological systems were extensively controlled during this period. In prehistoric times, the courses of both San Juan River, San Lorenzo River, and its tributaries were rerouted for 16.9 km (Fig 1) [23]. Despite the apparent need for agricultural production, Teotihuacanos elected to overlay many early irrigation systems with urban complexes [43–45]. Substantial labor and space were dedicated to performance of state-level ritualized activities in an elaborate ceremonial core constructed atop prime agricultural land along the valley bottom.

Researchers have interpreted Teotihuacan's urban landscape as a materialized cosmogram in which conceptions of time, space, history, and world order were conspicuously encoded in monumental structures [47–49]. The powerful projection of state ideological values through idiomatic architectural forms is said to be a main driver of Teotihuacan's expansion into one of the largest metropolitan centers of the New World [19, 47]. Its monumental construction presupposes the means to command substantial labor, logistics and material resources, and its ritual production necessitated commensurate participation of many city dwellers and foreigners in public spectacles (e.g. feasts, sacrificial ceremonies) [24, 50]. Separate tunneling operations at each pyramid revealed all three monuments to be compilations of bedrock mined from the Valley bottom augmented with stones and soil obtained from agricultural and cultural deposits [51]. Immense quantities of earth had to have been extracted from the surface to provide construction fill for the major buildings along the Avenue of the Dead, depriving the city of a convenient and potentially productive source of agricultural land [52]. Extensive excavations at Plaza of the Columns Complex, a centrally situated civic-administrative compound, allow us to directly correlate subsurface features with their topographic manifestations. Three-dimensional maps of the excavations are used to calculate volume of construction fill removed from the landscape.

This metropolis became a dominant regional power during its apogee (circa CE 250–550), with its influence recognized by the export of portable artifacts, architectural forms and the city's peculiar orientation [53, 54]. Even after its collapse circa CE 550, the legacy of Teotihuacan's influence continued both in the valley and abroad can be traced via this architectural tradition. Ongoing occupation (although in lower density) after the city's collapse meant that anthropogenic transformation of the valley's resources continued into the present, as subsequent populations repurposed materials from robust Teotihuacan-period structures.

## Landscapes as palimpsests of human action: A taskscape

Landscapes in the present trace the contours of their past: the palimpsest of layered material culture represents the accumulated record of ancient urbanism, from incipient formation of a state to its collapse and aftermath [55]. As Mlekuž [56] remarked in frustration, landscapes are inherently "messy", the cumulative superimposition of successive activities and temporalities are encoded physically (though to varying degrees) into the topography and environment of any given space and time [57–59]. An archaeological exploration reconstructs a landscape as an "artifact," a dynamic physical expression of cultural and natural processes with a biography that can be "read" through its physicality [60]. As human dwellers interact and engage with a landscape, their activity molds a "taskscape" into the solid physical environment [61]. Past decisions and actions thus often persist as topographic features layered onto the terrain. This insight can benefit decision makers contemplating the stewardship, transformation, or destruction of a given palimpsest landscape, as its alteration may have significant long-term consequences for modern communities.

In general, the uptick in interest for archaeological lidar have come from vegetated locations with high forest cover [62–65], sometimes with extant monocrop agricultural fields [66]. Initial publications show vivid visualizations and digital elevation models (DEMs) and tend to characterize this method as a revolutionary technology that helps peel back the forest canopy to reveal newly defined ancient city limits previously unrecognized from decades of survey work in a short period of time. However, additional work with lidar data has demonstrated the true intensive labor and time investments required in analyzing continuously resettled, urban, and environmental mosaics due to difficult feature definitions, issues with point cloud processing algorithms, and the overabundance of features, both ancient and modern, present on the landscape [56, 67–69]. Lidar data alone cannot resolve these 'messy' issues of temporality that can easily be misread and misinterpreted.

The strength and weakness of lidar is that they are indiscriminate snapshots of the totality of the topography of a given landscape during data collection. Many scholars have had to rely on a multi-methodological application that integrates satellite imagery, subsurface and surface survey, geomorphological techniques, and test excavations to disentangle deep-time cultural and natural legacies in each slope, mound, transect, and depression reported on the lidar map [57, 68, 70]. Lilley's [71] plan analysis of boundary walls on 19th and 20th century British maps traced these features to the Middle Ages, illuminating the propensity to fossilize underlaying urban landscapes, especially property demarcations. In fact, the east-west orientation of a major street continued to affect parallel or perpendicular ditch alignments even after the street/ditch was filled in. These mapping technologies visualize not only static three-dimensional landscapes, but also sculpt actions and reactions to the taskscape across the fourth dimension; time [72].

The challenge is quantifying the magnitude of human impact. Price et al. [1] maps degrees and extent of what they call "artificial grounds", land surfaces where human agency is

designated as prime modifiers. Archaeology shows once artificial grounds are created, they leave legacies as layers of cultural and natural transformations continue to superimpose on the active taskscape. Thus Price et al. [1] calculated up to 10m of artificial grounds in Manchester and Salford in northwestern England accumulated through continuous human intervention since the Roman period urban centralization. Inomata and colleagues [73] calculate 3,200,000–4,300,000 m$^3$ of earthen fill established the oldest monumental platforms in Mesoamerica, hidden until lidar mapping as seemingly natural ground surface at the Formative period site of Aguada Fenix (1000–800 BC). Geological maps designate artificial grounds [74] yet we argue archaeologists may be best suited to make these designations as a proper assessment of anthropogenic impact requires quantifying the scale of artificial ground (area and quantity of material moved), but also its rate (time span in which movement occurred) [1]. This study applies a combination of survey and excavation data to distinguish natural versus artificial ground with temporal and spatial control.

## The Teotihuacan lidar map as a geospatial database

The Teotihuacan lidar map was obtained in 2015 by the Project Plaza of the Columns Complex that covered an area of roughly 165 km$^2$, over 400% larger than Millon's original foot survey. Disentangling two thousand years of occupation on Teotihuacan's palimpsest landscape was the largest hurdle and most fulfilling discovery of the lidar mapping project. Though lidar point cloud processing algorithms often help peel away modern urban features [75], in our approach, we painstakingly recorded the orientation of modern features to realize how our daily movements are still molded by ancient subsurface features. We also conducted extensive ground reconnaissance and surface collection to map diachronic distribution of material culture across the landscape.

The primary project goals of the lidar mapping project were:

1. To refine and extend the coverage of the original survey map created by Millon and colleagues [29]. This pioneering survey recognized Teotihuacan's exceptional orthogonal city layout and scale, yet Millon's survey was conducted prior to the advent of lidar, and could not benefit from its greatly enhanced ability to incorporate detailed topographic shifts.

2. Given the full survey coverage of Teotihuacan's core by Millon's team, we can evaluate feature detection capacities of the two survey techniques in low-vegetation urban landscapes: conventional ground reconnaissance and lidar remote sensing.

3. To comprehend the total ecological footprint of Teotihuacan's occupation both spatially, outside of Millon's survey area, but also temporally; that is, the long-term impact of ancient landscapes onto the modern terrain.

4. Utilizing the lidar map as a digital archive of the Teotihuacan Valley to assess historical processes of urbanization (via comparison with historic survey data) and future (post-2015) artificial ground modifications.

Two projects provided robust survey data across the entire lidar area, and we have georeferenced many of their published survey plans [11, 12]. We are in the process of integrating the unpublished data from the Pennsylvania State archives. As the two projects were conducted nearly sixty years ago (TMP in 1962 and TVP in 1960), they provide critical historic documentation of the Teotihuacan Valley prior to large-scale mechanized agricultural practices. By coupling our lidar data with historic and current survey data, we documented complete loss of many previously recognized archaeological features. We also georeferenced a three-dimensional architectural map created through traditional total station mapping of every

consolidated and/or reconstructed architectural complex in the city as well as excavated areas accessible to the Moon Pyramid Project [76]. Stratigraphic control of not just Teotihuacan period architectural features, but the presence/absence of paleosoils and the height of the bedrock were critical to interpret how topographic features manifest subsurface features and for evaluating degree of artificial ground along key architectural units already explored archaeologically. The project continues to georeference published excavation drawings, salvage archaeology reports, and survey data to create a geospatial database to record as many extant and extinguished archaeological features.

## Results

The team identified five archaeological feature types (terraces, structures, mounds, plazas, and depressions) and recorded them as individual data layers in ArcGIS (Fig 3). In addition, two

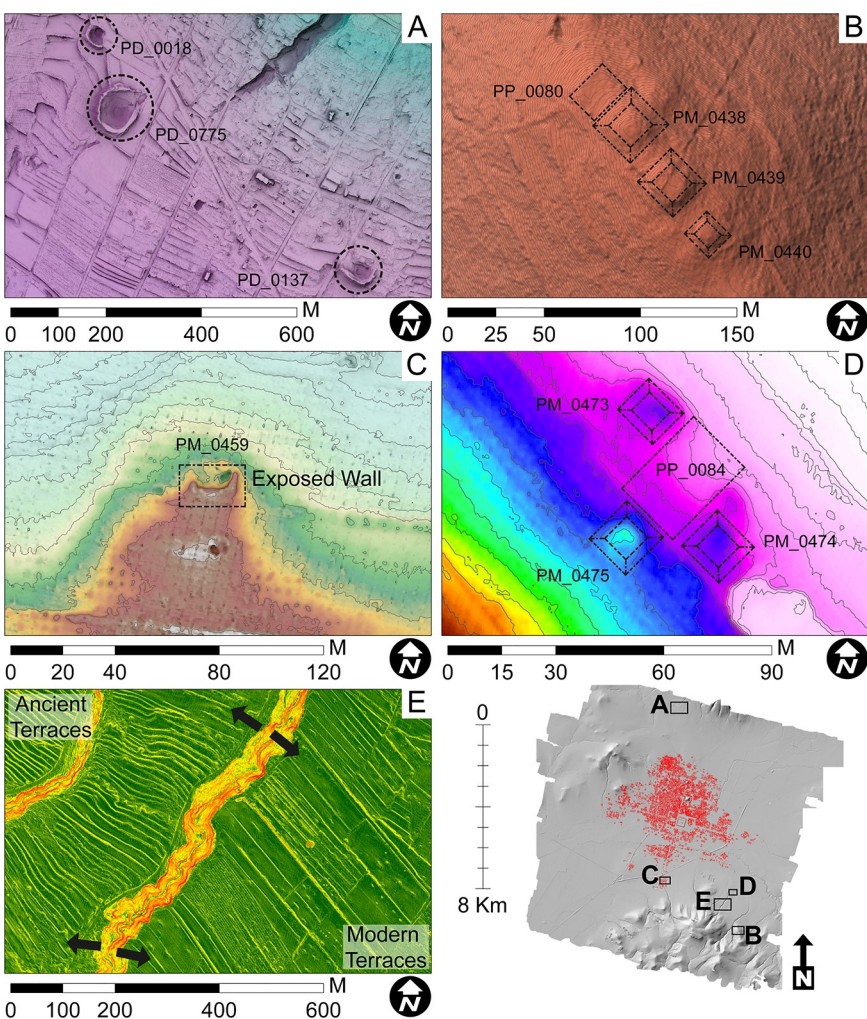

**Fig 3. Five feature types identified through the lidar map illustrating a variety of visualization methods.** A) depression (Skyview-factor), B) mounds (Hillshade Relief), C) structure (DEM with 0.5 m contour lines), D) mounds and a plaza (DEM with 0.5 m contour lines), and E) ancient and modern terraces (Slope Relief). Images produced with support by Ariel Texis. Published under a CC BY license, with permission from N. Sugiyama, original copyright Project Plaza of the Columns Complex 2021.

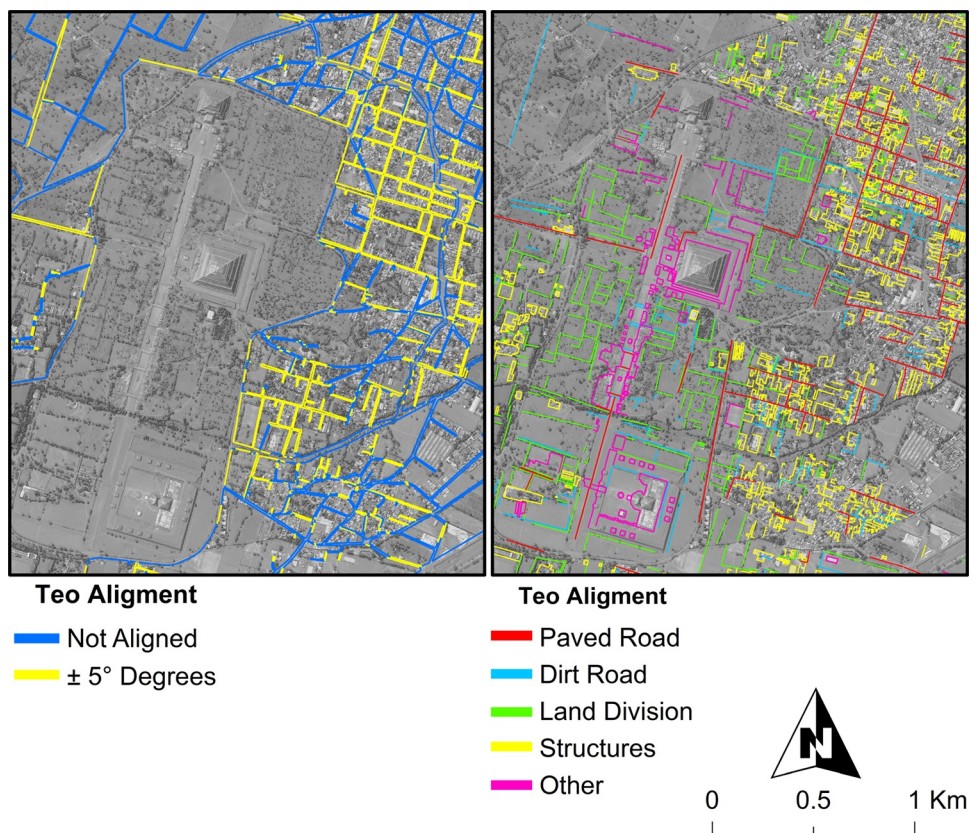

**Fig 4. Modern features within ±5° of the Teotihuacan alignments.** Left: land divisions. Right: other modern features. Satellite images acquired by Worldview-3, December 12, 2017, courtesy of Maxar Technologies. Published under a CC BY license, with permission from Maxar Technologies, original copyright Digital Globe Foundation 2018.

modern features that adhere to the Teotihuacan orientation were also mapped; Modern Property Alignments, which are manzana (land) divisions, and Modern Teotihuacan Alignments that represent modern paved/unpaved roads, boundaries, permanent structures, etc. (Fig 4). A first round of visual manual feature detection across the entire lidar map identified a total of 1,061 architectural features, 6,314 terraces, and 169 depressions (Table 1). The terraces, while most abundant, were more difficult to assess temporality but more accessible as they largely remain utilized as agricultural fields in the fringes of modern urbanization. 298 features (including depressions) and 5,795 terraces were identified outside of Millon's survey coverage (S1B Table in S1 File), many of which correlated with data from Sanders and colleagues [12], suggesting the broader urban imprint likely extended beyond Millon's map extent. This pattern was also reflected among Modern Teotihuacan Alignments and Modern Property Alignments, discussed below.

**Table 1. Summary of features recognized on the lidar map and later ground-verified (GV) (count, surface area, and percentage).**

| Feature Type | Total Count | Millon Map Inside/Outside | Arch Park Inside/Outside | Total Area | GV Count | GV Area | % Area GV |
|---|---|---|---|---|---|---|---|
| Mounds | 730 | 592/138 | 349/381 | 0.53 km² | 637 | 0.45 km² | 85% |
| Structures | 264 | 250/14 | 173/91 | 0.50 km² | 242 | 0.46 km² | 92% |
| Plazas | 67 | 57/10 | 49/18 | 0.27 km² | 61 | 0.26 km² | 96% |
| Depressions | 169 | 33/136 | 0/169 | 0.35 km² | 75 | 0.26 km² | 74% |
| Terraces | 6314 | 519/5,795 | 0/6,314 | 6.2 km² | 3,027 | 3.28 km² | 53% |

**Table 2. Feature accuracy based on pre- and post-ground verification (GV) assessment by feature type.**

| Feature Type | Confidence Level 0 | | | Confidence Level 1 | | | Confidence Level 2 | | | Confidence Level 3 | | |
|---|---|---|---|---|---|---|---|---|---|---|---|---|
| | Count | Post GV 3 | Accuracy | Count | Post GV 3 | Accuracy | Count | Post GV 3 | Accuracy | Count | Post GV 3 | Accuracy |
| Mounds | 1 | 1 | 100% | 82 | 38 | 46% | 161 | 102 | 63% | 393 | 393 | 100% |
| Structures | 3 | 1 | 33% | 22 | 13 | 59% | 30 | 16 | 53% | 187 | 187 | 100% |
| Plazas | 0 | 0 | - | 2 | 0 | 0% | 6 | 2 | 33% | 53 | 52 | 98% |
| Depressions | 0 | 0 | - | 22 | 1 | 5% | 45 | 16 | 36% | 8 | 3 | 38% |
| Terraces | 720 | 116 | 16% | 774 | 114 | 15% | 1,025 | 188 | 18% | 508 | 103 | 20% |

Only those considered confidence level 3 post-GV are considered "accurate".

Many locals where Millon's team had delineated structures had no corresponding topographic indicators of subsurface features. Many Modern Teotihuacan Alignments followed the edges of Millon's potential structures, suggesting that the survey crew relied heavily on both artifact scatter and modern alignments for his hypothetical structure loci and orientation.

Specific excavation and architectural data mapped with a total station was added onto the lidar map to embed stratigraphic information where appropriate. For example, the bedrock level along key excavations at Plaza of the Columns Complex and within the tunnels of the Sun Pyramid and Moon Pyramid were essential for volume calculations of artificial ground. Stratigraphic data, such as the lack of paleosoil layers, were key to identify areas where anthropogenic alterations have modified the bedrock surface.

Ground verification survey of roughly 1,300 features spanning 10.3 km$^2$, and a total walked area (including areas where no features were found) of 13.3 km$^2$ was completed between 2017–2019 survey seasons. Each feature was assigned an initial confidence level, on a scale of 1 to 3, indicating whether they were more likely to be modern/natural (1) or ancient (3). The field crew would then assign each feature a second confidence level post ground reconnaissance. Feature detection accuracy was assessed by comparing initial confidence level to its designation post-ground verification (Table 2). Confidence level 3 accuracy was quite precise (usually above 90%), with the exception of depressions and terraces that are notoriously difficult to date. Confidence levels 1 and 2, on the other hand, require chronological verification (roughly 30–60%). Confidence level 0 features represent omission errors of features detectable only via ground reconnaissance, even after a re-evaluation of the lidar DEM. Roughly 83% (1015/1230) off all features except terraces (50%) were ground verified. This high number is largely attributable to the high density of features within the Teotihuacan Archaeological Park, which we have extensively mapped prior to this survey project that were considered "ground verified". Due to the low accuracy in confidence level 1 and 2 features outside of Millon's survey area, we conducted targeted sampling to define structures (80% ground verified), plazas (71%), mounds (60%), and terraces (46%), while depressions (26%) were both more difficult to date despite ground verification, and also substantially more abundant (S1C Table in S1 File). In total, we have completed 46% coverage of confidence level 1 and 2 features outside of Millon's survey area.

## Discussion

### Modern alignments

Heat maps generated of the total length of Modern Property Alignments (Fig 5A) and Modern Teotihuacan Alignments (Fig 5B) visually maps localities where Teotihuacan's orthogonal city plan continues to influence daily interaction with the landscape (Fig 5C). Roughly 29% of the

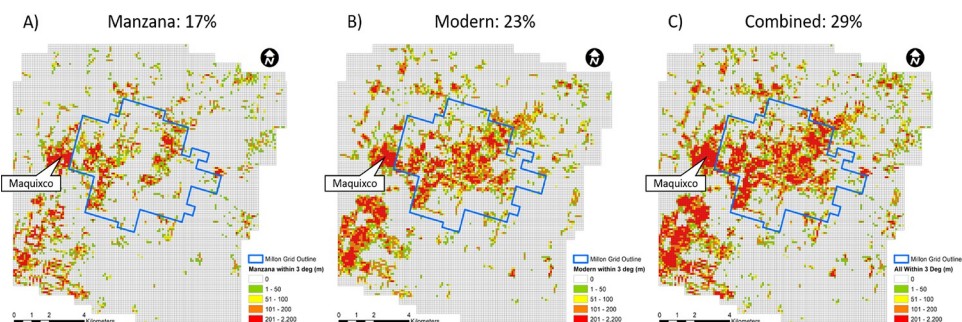

**Fig 5. Distribution heat maps.** A) Modern Property Alignment (Manzana boundaries), B) Modern Teotihuacan Alignment, and C) combined Teotihuacan Alignment features. Map displays total length for each 100m² grid and % of grids with values above 1. Map produced by T. Catignani.

entire lidar coverage area and 65% of urban areas contained property or modern features that aligned within three degrees of 15˚east of astronomical north (S1D and S1E Table in S1 File).

Excavations at Plaza of the Columns Complex targeted several modern rock piles defined as modern Teotihuacan alignments. We determined they delineate large perimeter walls and roomblocks usually too thick for manual plows to penetrate (Fig 6). Ancient subsurface structures like perimeter walls, buildings and mounds—whether negatively as obstacles to be avoided or positively as attractive caches of worked and dressed stone—continually informs the geometry of modern construction, including pipeline routing and surface grading.

The high concentration of Teotihuacan aligned features in the town of Maquixco, for example was surprising given its distance from the city's core (Fig 5). Later, we found Sanders had mapped Teotihuacan features in the area, that were later excavated as TC-8 [33, 46]. The georeferenced survey maps helped us find several mounds that were originally unrecognized on the lidar map, and still others that have completely vanished due to extensive cultivation in the area. The extremely high density of Teotihuacan alignment in the rest of Maquixco township suggests perhaps other Teotihuacan period occupations may still lay below the surface. The distribution of Teotihuacan alignments into areas well outside the city center suggests the impact of the ancient urban city on the modern terrain extends well beyond Millon's map area. We suspect more excavation projects, especially integrating salvage archaeology operations, can help expand the definition of the grand metropolis.

## Artificial ground calculations

The Teotihuacan Valley is constantly under construction and deconstruction of expansive artificial ground. Here we highlight some of the most conspicuous examples, but as more excavation data is gathered into the three-dimensional database, we will be able to more precisely calculate the volume of artificial ground across the valley. Perhaps, the most obvious of these is the sheer volume of the three pyramidal complexes (2,423,411 m³ volume) that is entirely comprised of bedrock, stones, adobes, and soil utilized as construction debris scrapped from mining the Valley bottom (Table 3). This volume calculation does not include the platforms that comprise the Moon Plaza Complex and the parameter wall surrounding the Sun Pyramid, which would further enlarge their volumes. Interestingly, these volume estimates based on combining high precision lidar surface models with AutoCAD excavation data of the bedrock elevation provided volume estimates substantially distinct from that reported in the literature [52, 77, 78]. The table also provides a precise volume estimate of the combined value of the

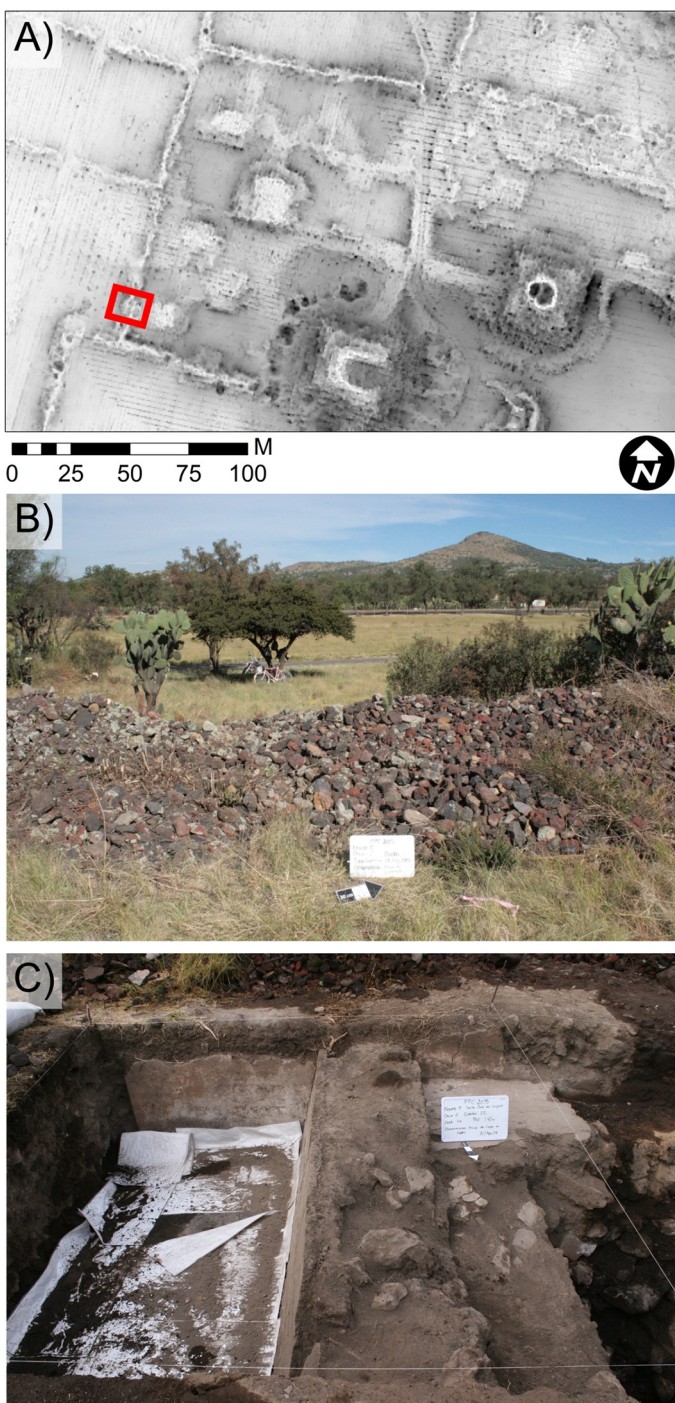

**Fig 6. Images pertaining a modern rock pile.** A) Sky-view factor image of Plaza of the Columns showing a modern rockpile delineating the complex, rectangle indicates excavation unit along a Teo Modern alignment, B) photograph of the modern rock pile, C) photograph of excavations revealing large wall beneath modern rock pile. Published under a CC BY license, with permission from N. Sugiyama, original copyright Project Plaza of the Columns Complex 2021.

Feathered Serpent Pyramid, the large perimeter wall of the Ciudadela, and structures enclosed within the complex that were significantly higher in volume than the Moon Pyramid's construction. Barba and Frunz [52] calculates the 18 bedrock quarries throughout the valley

**Table 3. Volume calculations of the Moon Pyramid, Sun Pyramid, and Ciudadela/Feathered Serpent Pyramid.**

| | Prior Volume Calculations (m³) | | | PPCC Volume Calculation |
|---|---|---|---|---|
| | Barba and Frunz 2010 | Murakami 2010 | Millon 1960 | |
| Sun Pyramid | 1,269,611 | 1,234,046 | 944,000 | 1,249,937 |
| Moon Pyramid | 329,333 | 227,229 | 228,600 | 412,078 |
| Ciudadela | - | - | - | 761,396 |
| PPCC | - | - | - | 372,056 |

produced nearly 670 million m³ of bedrock and tuff. They argue this accounts for 35% of the volume of the principal structures at the ceremonial core. Yet immense volume of bedrock to build the city remains unaccounted, as each floor and wall heavily relied on bedrock to level the floors and tuff as building blocks. Large portions of the valley bottom were subject to extensive alteration that do not leave obvious traces as quarries as they lay deep below occupation layers.

A pedogenic study of paleosols in the Teotihuacan Valley sampled fill of the Moon Pyramid to find it was constructed through large-scale destruction of paleosols despite its importance as an agricultural resource rich in nutrients [79]. Deep stratigraphic data from the Plaza of the Columns Complex excavations demonstrated the majority of the pits had no evidence of paleosols as the bedrock of the entire area was artificially flattened for the dual purpose of architectural stability and to extract valuable bedrock, eliminating the paleosol layer. Of the 272 pits and nine tunnels excavated in this complex over the course of four field seasons, only one (Front F) had evidence of paleosols, with 3D topographic maps recording the height of the original bedrock layers (Fig 7). Many areas showed they were intentionally flattened for constructing plazas or walls. We calculate approximately 372,056 m³ of artificial ground accumulated over the course of roughly 300 years (CE 150–450) of primary construction activity at Plaza of the Columns Complex that must have been quarried from the Teotihuacan Valley (Fig 8). This is a volume that is just under volume of the Moon Pyramid construction (412,078 m³) for this pair of complexes alone. Though the bedrock elevation usually stayed within expected degree of variation (especially flat around plazas where they were leveled), deep cuts in the bedrock recorded in some areas (Area 1, Fig 8) suggests systematic quarrying along the

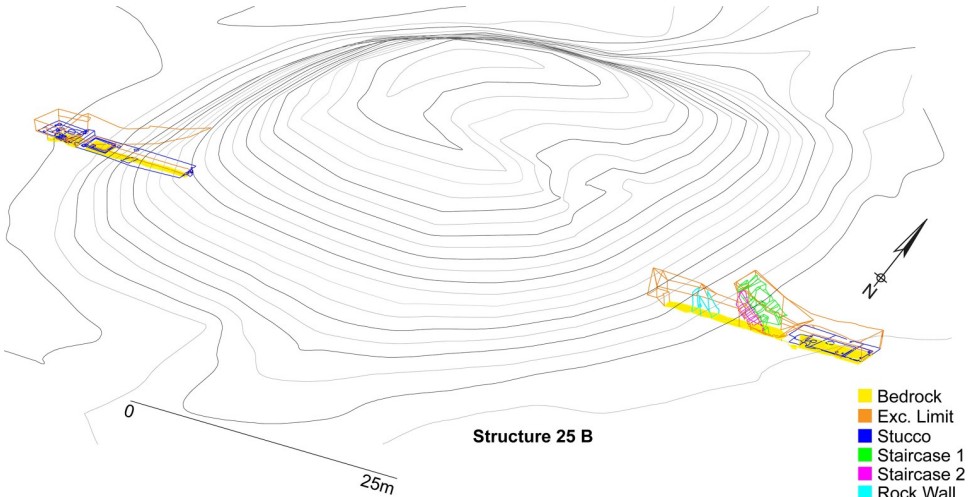

**Fig 7. 3D AutoCAD drawing of trench and tunnel excavations of structure 25B indicating height of bedrock (yellow).** Published under a CC BY license, with permission from S. Sugiyama, original copyright Moon Pyramid Project 2021.

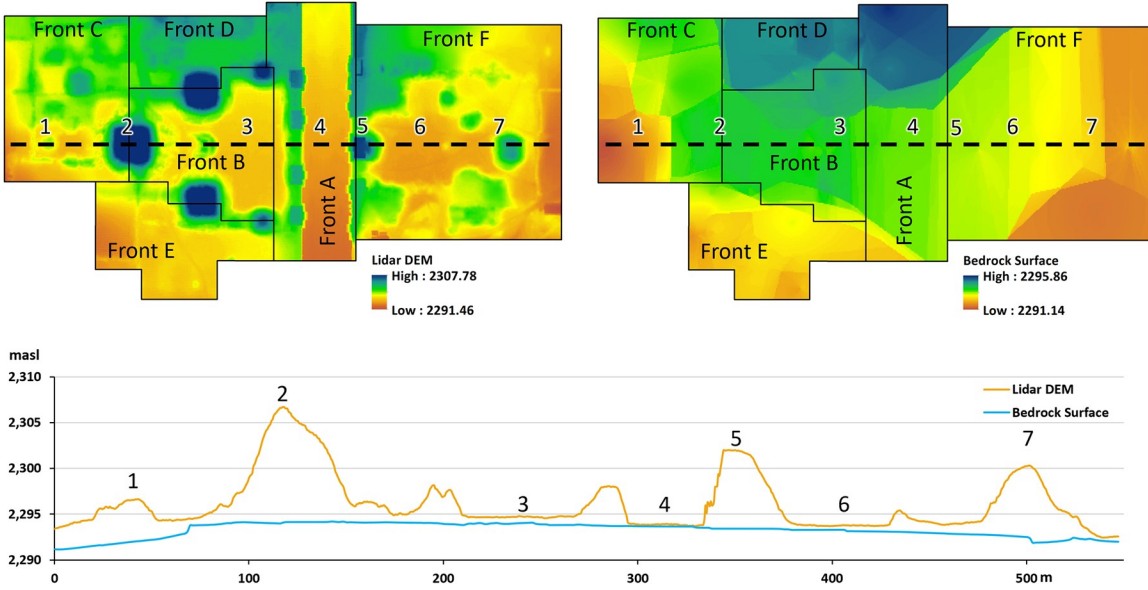

**Fig 8. Artificial ground volume calculation methodology.** Upper left: Lidar DEM indicating the topography (height in masl). Upper right: Bedrock surface model based on excavations at Plaza of the Columns Complex. Below. Profile of #1 and #2 used to calculate volume. Note abrupt cut in bedrock in region 1. Map produced by T. Catignani.

periphery of the complex. The topography of most of the ceremonial center is also likely altered down to its bedrock. Considering similar sized complexes align the length the Avenue of the Dead with comparable deep stratigraphic profiles, our intention is to amplify volume calculations to the rest of the ceremonial core of Teotihuacan through the combination of AutoCAD mapping of reconstructed and excavated zones, as well as georeferencing previous excavation drawings onto the map into the future. We are just scratching the surface in defining the scale of landscape alterations at this massive cityscape.

Though such volume calculations within the Teotihuacan Valley at large would require more stratigraphic data from subsurface structures throughout the Teotihuacan Valley, we have recognized modern (60%, though likely mixed), ancient (22%), and mixed (3%) artificial ground areas comprised the majority of the lidar map, with natural zones only totaling 15% of the 165 km² area (Fig 9).

Another substantial feat of human landscape alteration was their investment in redirecting the two major rivers that traverse the city, the Rio San Juan and the San Lorenzo River. S. Sugiyama [47] interpreted them as major canals of symbolic and calendric significance based on his measurement unit study. The former follows the Teotihuacan orientation for a total 3 km as it traverses the city center (first E-W, then N-S), and the latter has a very distinct orientation, 8° south of astronomical east for 4.9 km (Fig 9). The lidar map also reveals that other sections of canals and rivers, many still actively utilized today, were altered at various points along its course, frequently coinciding with the Teotihuacan directionalities (Fig 9, blue lines). A total of 16.9 km of the hydrological systems visible on the modern terrain had origins in the Early Classic Teotihuacan landscape.

## Lidar applications to urban taskscapes

After nearly four years of feature detection and ground survey, we can say that lidar applications in palimpsest taskscapes are significantly more time consuming, labor intensive, and

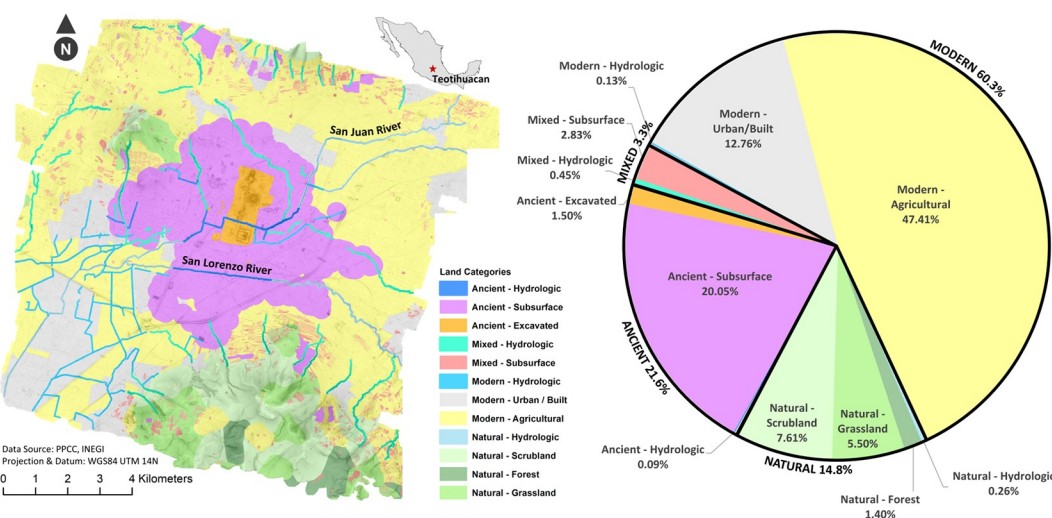

**Fig 9. Distribution of natural versus modern, ancient, and mixed artificial ground calculated from the lidar map of the Teotihuacan Valley.** Total area represents 165 km$^2$. See S1F Table in S1 File for details of surface area.

requires much more horizonal (survey) and vertical (excavation) depth to interpret. Lidar technology cannot substitute traditional pedestrian survey, as both DTMs nor DEMs are never true proxies of ground surface nor a perfect reflection of all landscape alterations. The distribution of features detected through the lidar map significantly under-represent those proposed through traditional survey (31% in count and 10% in spatial overlap). Our results parallel Prufer and colleague's [67] assessment in disturbed tropical environments of Belize at Uxbenká, both modern agricultural practices and mosaic nature of the environment hindered feature detection, especially of insubstantial structures. Archaeological features lower than a meter or meter and a half were often unrecognizable on the DEM at first glance, detectable only after adding contour lines at the 0.5 m intervals after ground verification (Fig 3C and 3D). Modern machine-based agricultural practices have eradicated many mounded features and terrace systems (Fig 3E) originally reported in historical survey records.

Despite these challenges, the Teotihuacan lidar map has been able to register 298 architectural features and 5,573 terraces outside of Millon's map (many of which were previously undocumented), refine several of the features within Millon's survey extent, and has documented the detrimental effects of recent urbanism and machine-operated artificial ground modifications that have substantially erased many features extant during pioneering surveys in the 1960s. We have recorded 1.5 km$^2$ of mined area since 2015 have vanquished 192 terraces, 7 mounds, and 6 depressions, many of which were never reported in previous surveys (S1A Table in S1 File).

## Conclusions and future directions

For the purposes of this paper we report the features on the Teotihuacan lidar map as they are, as a 'messy' layer of an active culmination of long-term human-landscape interaction with visible consequences of urban taskscapes that still structure our perception and experience on the ground. This map confirms the palimpsest nature of the Teotihuacan valley. Many of the subsurface archaeological features continue to influence modern land divisions and road alignments outside the city center that still adhere to this distinctive orientation. In fact, the modern terrain provides important clues of the subsurface features.

The broader impact of this map is already clear. The original lidar data taken in 2015 serves as a snapshot of the Teotihuacan landscape, preserving archaeological features as a digital archive

erased by urban growth. Our lidar map is already the only historical record of many archaeological features no longer available for study and can become a method to monitor the extent of landscape alterations into the future. This project follows Parcak's [80] initiative that the only way to preserve many of the archaeological features is via digital archiving and monitoring of heritage landscapes as we are increasingly overwhelmed by the immediate urbanization and resource crisis we face that affect vast areas with unprecedented speed across the globe.

The lidar map has become the basis to create a three-dimensional geospatial database, in which to overlay stratigraphic and surface data, define artificial and natural ground surfaces, and to document the true extent of humans as geomorphic agents of the Teotihuacan Valley. The lidar map will continue to evolve with the dynamic landscape, as the project's second phase is focusing on georeferencing more historical survey data and excavation data that are non-reproduceable from the modern terrain. As responsible stewards of this data, we are currently beginning a long-term collaboration with the Salvage Archaeology sector of the Mexican National Institute of Anthropology and History, which will facilitate recognizing ancient features and rescue operations as irreplaceable archeological heritage is subducted under the sprawling march of modern urban expansion. Ethical guidelines for lidar data preservation and dissemination are still being developed within archaeology [81, 82]. Our project will continue to work with Mexico's National Institute of Anthropology and History to effectively integrate appropriate stakeholders.

Archaeologists are uniquely trained to evaluate extent and long-term consequences of anthropogenic alterations at a landscape scale. We have recorded massive terrace systems along entire mountain ranges (e.g. Maya and Inca) [83, 84] and recognized modern effects of ancient cultures moving large bodies of soil for agriculture and construction (e.g. Europe and Amazon) [1, 85]. Understanding urban landscapes through the *long-durée* can help understand the legacies of ancient cities.

## Materials and methods

### Data acquisition and feature detection

The Project Plaza of the Columns Complex has complied with all relevant regulations and necessary permits for lidar point capture and ground reconnaissance were obtained from the National Institute of Anthropology and History (INAH) from 2015–2019 (Oficios 401.B(4) 19.2015/361174, 401.B(4)19.2016/36/1024, 401.1S.3-2017/1031, 401.1S.3-2018/963, 401.1S.3-2019/1058). The National Center for Airborne Laser Mapping (NCALM, University of Houston-Texas) captured the lidar dataset in 2015 using the Titan MW system [86]. Additional processing for sky-view factor and slope model visualizations proved useful for analysis. S2 File details data acquisition parameters and visualization methodology.

Lidar-based feature identification was accomplished by first comparing several lidar-derived visualizations, satellite imagery, and historic survey reports. Five feature types (terraces, structures, mounds, plazas, and depressions) were captured as individual data layers in ArcGIS. Alongside archaeological features Modern Property Alignments and Modern Teotihuacan Alignments with Teotihuacan orientations were defined using CONABIO [87] and SEDATU [88] land division (manzana) data, as well as satellite imagery. Any feature oriented 15˚±3 northeast of astronomical north was considered a Teotihuacan alignment, though we tested the results as various accuracy intervals up to 5 degrees of accuracy. This accounted for inter-analysis errors and mounding effects of subsurface structures to manifest their original alignments on the modern terrain.

We devised a year-round feedback system between the feature detection team (in the United States) interacting intensively with the ground verification team (in Mexico) that

encourage continued learning and improvement on behalf of both the feature detection and ground reconnaissance team. Several innovative methods applied include ArcGIS Online, ESRI Collector application accessed through project ipads, and WhatsApp application for continuous feedback. A brief description of each feature and more detail on feature detection/calculation methodology is reported in S2 File.

## Artificial ground volume calculations

Precise and systematic documentation of the bedrock elevation from total station mapping of the Project Plaza of the Columns Complex excavations (Fig 7, yellow polygon) was used to create a bedrock elevation model [see also 73]. This bedrock elevation model was compared to the surface DEM derived from our lidar model to calculate the volume (Fig 8). More details on modeling a bedrock surface elevation model from these bedrock heights and total volume calculations were obtained in ArcGIS and Surfer and can be consulted in S2 File.

## Artificial ground area calculations

We divided the project area into four categories of land type: 1) Artificial–Ancient, 2) Artificial-Mixed, 3) Artificial-Modern, and 4) Natural. The first category consists of all lidar-detected features that have a confidence level of 3 for both pre- and post-ground verification, the entire archaeological park, all excavated areas, and canals that were modified by Teotihuacan's ancient inhabitants. Millon's [29] criteria of 300 m border around all structures was applied to delineate ancient subsurface features within his survey area. The second category consists of lidar-detected features with a confidence level of 2 or 1 for post-ground verification, and canals that have been used in both ancient and modern times. The third category includes all remaining lidar-detected features, modern canals, and modern agricultural areas. The fourth category represents the space within the study area that does not appear to have been modified by ancient or modern inhabitants. Natural land categories assigned by INEGI were integrated into the map [89]. For each of these categories, we dissolved the data into separate layers and calculated the total area of each, ensuring that total was equal to the lidar survey area and that no areas were counted twice.

## Ground reconnaissance

Ground reconnaissance was essential to evaluate feature accuracy. We targeted 50% coverage of each feature type with accuracy of one or two for areas outside of Millon's survey area, excluding terraces that were simply too abundant and often do not result in clear temporal designation even after ground verification. Hand-held GPS trackers were used to record the locality of a 10x10m surface collection track that will help further assign distribution of artifacts, especially ceramic phases across the surveyed area. Survey data comparable to Millon's data forms (11:Fig 11) were captured into a spreadsheet. We are still analyzing the surface collections. Post ground-verification accuracy was assigned based on the presence of architectural features visible on the surface, the degree and characteristics (e.g. ceramic type) of the surface collection. Consulting local land owners aided documenting historical usage of the landscape, which was also noted in our survey forms.

## Supporting information

**S1 File. Additional tables.**
(PDF)

**S2 File. Methodologies.**
(PDF)

## Acknowledgments

The National Center for Airborne Laser Mapping (NCALM, University of Houston-Texas) completed the lidar data acquisition, and we would like to especially thank Ramesh L. Shrestha for all the technical support throughout this project. Excavation and survey work were conducted with permission from the National Institute of Anthropology and History. Project Plaza of the Columns Complex is co-directed by S. Sugiyama, N. Sugiyama, V. Ortega, W. Fash, and D. Carballo. Lidar ground reconnaissance was completed by Ariel Texis and Omar Rodríguez. Lidar feature detection was completed with the support of many students. We would like to thank Ariel Texis for support making some of the graphics utilized in this article.

## Author Contributions

**Conceptualization:** Nawa Sugiyama, Saburo Sugiyama.

**Data curation:** Nawa Sugiyama, Saburo Sugiyama, Juan C. Fernandez-Diaz.

**Formal analysis:** Nawa Sugiyama, Saburo Sugiyama, Tanya Catignani, Adrian S. Z. Chase, Juan C. Fernandez-Diaz.

**Funding acquisition:** Nawa Sugiyama, Saburo Sugiyama.

**Project administration:** Nawa Sugiyama, Saburo Sugiyama.

**Supervision:** Nawa Sugiyama.

**Visualization:** Tanya Catignani.

**Writing – original draft:** Nawa Sugiyama, Tanya Catignani, Adrian S. Z. Chase, Juan C. Fernandez-Diaz.

**Writing – review & editing:** Nawa Sugiyama, Saburo Sugiyama, Tanya Catignani, Adrian S. Z. Chase, Juan C. Fernandez-Diaz.

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
