## [Decision Letter · Decision Letter 0]

27 Apr 2021

PONE-D-21-08331

Lidar detection of humans as geomorphic agents: The past, present and future of the Teotihuacan Valley, Mexico

PLOS ONE

Dear Dr. Sugiyama,

Thank you for submitting your manuscript to PLOS ONE. After careful consideration, we feel that it has merit but does not fully meet PLOS ONE’s publication criteria as it currently stands. Therefore, we invite you to submit a revised version of the manuscript that addresses the points raised during the review process.

All comments need to be addressed before re-submission.

We look forward to receiving your revised manuscript.

Kind regards,

Peter F. Biehl, PhD

Academic Editor

PLOS ONE

Journal Requirements:

3. In your manuscript, please provide additional information regarding the specimens used in your study. Ensure that you have reported specimen numbers and complete repository information, including museum name and geographic location.

For more information on PLOS ONE's requirements for paleontology and archaeology research, see https://journals.plos.org/plosone/s/submission-guidelines#loc-paleontology-and-archaeology-research

5. We note that Figures S2 a-d, Figure 1, 2, 3, 4, 5, 6 and 9 in your submission contain map/satellite images which may be copyrighted.

a. You may seek permission from the original copyright holder of Figures S2 a-d, Figure 1, 2, 3, 4, 5, 6 and 9  to publish the content specifically under the CC BY 4.0 license. 

Additional Editor Comments:

Please address all comments before re-submission.

Reviewers' comments:

Reviewer's Responses to Questions

**Comments to the Author**

1. Is the manuscript technically sound, and do the data support the conclusions?

Reviewer #1: Yes

Reviewer #2: Yes

2. Has the statistical analysis been performed appropriately and rigorously? 

Reviewer #1: Yes

Reviewer #2: Yes

3. Have the authors made all data underlying the findings in their manuscript fully available?

Reviewer #1: Yes

Reviewer #2: No

4. Is the manuscript presented in an intelligible fashion and written in standard English?

Reviewer #1: Yes

Reviewer #2: Yes

5. Review Comments to the Author

Reviewer #1: The paper is a valuable addition to literature on Teotihuacan by enlarging the previously mapped area by over four times in extent with digital technologies that allow for the identification of a larger range of features, higher precision of the form and location of features, and identification of changes in the anthropogenic landscape over time. The last is primarily how the authors frame the study, in that the new dataset can help to understand the loss of segments of the archaeological record at this globally important site and hopes for conservation efforts in the future. Another interesting facet of the project is its documentation of “path dependence,” sometimes in a literal sense, of the alignments of features of the built environment conforming to the orthogonal layout of the Classic period city. As the authors propose, this could be a way of identifying possible hamlets of that era elsewhere in the Teotihuacan Valley that were overlaid by Aztec period, Colonial, and or contemporary sites. New calculations of cubic meters of construction for the three major temple complexes plus Plaza of the Columns Complex are also very useful contributions to the literature.

I suggest the following minor revisions to the text, which are primarily about wording except for the final point about calculating construction volumes.

The current title is phrased awkwardly, as the Lidar detection is not of humans, rather of remains they created. Rewording to something such as “Humans as Geomorphic Agents: Lidar Detection of the Past, Present, and Future of the Teotihuacan Valley, Mexico” makes more sense.

32 Abstract line 1 – remove the dangling participle (humans are the primary geomorphic agents on the landscape, not human-induced artificial ground)

34 Abstract line 2 – change repetition of word landscapes

113 – put the degree sign after 15.5

169 – since the state had collapsed, I would clarify “the legacy of the Teotihuacan’s influence continued both in…”

292 Table 3 and elsewhere – since the term Ciudadela is so problematic—the structure is not actually a citadel—I would suggest writing it out using the Spanish label (Ciudadela) or branding as Feathered Serpent Pyramid Complex, rather than Citadel.

404-406 – One caution in the comparisons between the Moon Pyramid and the PCC (or the entire Ciudadela) is that the first is just calculating the volume of the monument, while the second considers the complex as a whole, including basal platform and structures on top of it. For more precise comparison the Moon Pyramid should also be considered as a complex with the volume of the two attached flanking platforms (6:N5W1 and 22:N5E1) included.

Reviewer #2: There has been little application of LiDAR to archaeological research in the central highlands of Mexico. This research is a valuable application for assessing the utility of this method to detect prehispanic archaeological features. Their finding that the lidar map significantly under-represented features detected through pedestrian archaeological survey. Central Mexico, at least prior, to explosive urban growth adn construction that begin in the mid-twentieth century offered excellent conditions for traditional ground survey. Nonetheless this application of lidar produced valuable results includiing teh need to expand Millon's map of Teotihuacan, seeing how remains of the 2000 year old city have and continue to structure the historic and modern landscape and developing a geospatial data base combining data remote sensing and conventional archaeological survey and mapping.

Recognizing the valuable work of hte researchers, the impact of this research will rest on making their data accessible so that it can be built on by other researchers including Mexican archaeologists responsible for them.

p. 19 The authors should clarify if they consulted Sanders survey data records at Penn State that contain significant information not in the published volumes from excavations and surveys of sites.

6. PLOS authors have the option to publish the peer review history of their article (what does this mean?). If published, this will include your full peer review and any attached files.

Reviewer #1: No

Reviewer #2: No

---

## [Author Response · Author response to Decision Letter 0]

16 Jul 2021

Reviewer #1 Comments: 

We thank Reviewer #1 for their positive review and minor edits and suggestion. All wording comments have been addressed as indicated by the change tracking system. Below we elaborate on how we have taken into consideration each point. 

1. Title change suggestion from “Lidar detection of humans as geomorphic agents: The past, present and future of the Teotihuacan Valley, Mexico” to “Humans as Geomorphic Agents: Lidar Detection of the Past, Present, and Future of the Teotihuacan Valley, Mexico” has been made as suggested by the reviewer.

2. Reviewer requested we remove the dangling participle. We have changed sentence to “As humans are the primary geomorphic agents on the landscape, it is essential to assess the magnitude, chronological span, and future effects of artificial ground that is expanding under modern urbanization at an alarming rate.”

3. Reviewer has suggested we change repetition of word landscapes. We have changed the text to, “We argue humans have been primary geomorphic agents of landscapes since the rise of early urbanism that continue to structure our everyday lives.”

4. We have put the degree sign after 15.5.

5. Since the state had collapsed, we have clarified the sentence to “the legacy of the Teotihuacan’s influence continued both in…” as requested by the reviewer. 

6. Table 3 and elsewhere – The reviewer mentions that because the term Ciudadela is so problematic—the structure is not actually a citadel— he/she suggested using the Spanish label (Ciudadela) rather than Citadel. We have fixed all mention of the Citadel to Ciudadela in the text and figures. 

7. The only content-related suggestion was a cautionary note about comparing the Moon Pyramid volume with the Plaza of the Columns Complex volume because the former is a monument while the latter is an entire complex including basal platforms and residential complexes. Interestingly, despite this, Plaza of the Columns Complex volume is still lower than the Moon Pyramid. 

The reason we have not included other flanking platforms is because we only have bedrock layer information for the Moon Pyramid itself, while our excavations at the Plaza of the Columns provided bedrock data for the entire complex. In addition, we cite the existing literature of volume calculations, which only provided volume estimates for the Moon Pyramid itself. Our hope is to create a volume estimate for the entire complex as we expand our database of known bedrock elevations. As stated in the text, these volume calculations are the beginning of a more accurate volume calculation of the ceremonial core at large, as we state, “Here we highlight some of the most conspicuous examples, but as more excavation data is gathered into the three-dimensional database, we will be able to more precisely calculate the volume of artificial ground across the valley.”

To avoid implying that these volume estimates are directly comparable, we have added the sentence, “This volume calculation does not include the platforms that comprise the Moon Plaza Complex and the parameter wall surrounding the Sun Pyramid, which would further enlarge their volumes.”

Reviewer #2 Comments

His/her concise and positive review only requested clarification concerning whether the Sanders survey data records of unpublished materials at Penn State were integrated into our map. We have modified the text to satisfy this clarification, “Two projects provided robust survey data across the entire lidar area, and we have georeferenced many of their published survey plans [11,12]. We are in the process of integrating the unpublished data from the Pennsylvania State archives.”

---

## [Decision Letter · Decision Letter 1]

6 Sep 2021

Humans as geomorphic agents: Lidar detection of the past, present and future of the Teotihuacan Valley, Mexico

PONE-D-21-08331R1

Dear Dr. Sugiyama,

We’re pleased to inform you that your manuscript has been judged scientifically suitable for publication and will be formally accepted for publication once it meets all outstanding technical requirements.

Kind regards,

Peter F. Biehl, PhD

Academic Editor

PLOS ONE

Additional Editor Comments (optional):

Reviewers' comments:

Reviewer's Responses to Questions

**Comments to the Author**

1. If the authors have adequately addressed your comments raised in a previous round of review and you feel that this manuscript is now acceptable for publication, you may indicate that here to bypass the “Comments to the Author” section, enter your conflict of interest statement in the “Confidential to Editor” section, and submit your "Accept" recommendation.

Reviewer #1: All comments have been addressed

2. Is the manuscript technically sound, and do the data support the conclusions?

Reviewer #1: (No Response)

3. Has the statistical analysis been performed appropriately and rigorously? 

Reviewer #1: (No Response)

4. Have the authors made all data underlying the findings in their manuscript fully available?

Reviewer #1: (No Response)

5. Is the manuscript presented in an intelligible fashion and written in standard English?

Reviewer #1: (No Response)

6. Review Comments to the Author

Reviewer #1: (No Response)

7. PLOS authors have the option to publish the peer review history of their article (what does this mean?). If published, this will include your full peer review and any attached files.

Reviewer #1: No

---

## [Editor Report · Acceptance letter]

10 Sep 2021

PONE-D-21-08331R1 

Humans as geomorphic agents: Lidar Detection of the past, present and future of the Teotihuacan Valley, Mexico 

Dear Dr. Sugiyama:

I'm pleased to inform you that your manuscript has been deemed suitable for publication in PLOS ONE. Congratulations! Your manuscript is now with our production department. 

Kind regards, 

on behalf of

Dr. Peter F. Biehl 

Academic Editor

PLOS ONE